

# Characterization of LC-MS based urine metabolomics in healthy children and adults

Xiaoyan Liu[1,*], Xiaoyi Tian[2,3,*], Shi Qinghong[4], Haidan Sun[1], Li Jing[1], Xiaoyue Tang[1], Zhengguang Guo[1], Ying Liu[2], Yan Wang[2], Jie Ma[2], Ren Na[2], Chengyan He[4], Wenqi Song[2,3] and Wei Sun[1]

[1] Proteomics Research Center, Institute of Basic Medical Sciences, Chinese Academy of Medical Sciences, School of Basic Medicine, Peking Union Medical College, Beijing, China

[2] Department of Clinical Laboratory, National Center for Children's Health, Beijing Children's Hospital, Capital Medical University, Beijing, China

[3] Beijing Advanced Innovation Center for Big Data-Based Precision Medicine, Beihang University & Capital Medical University, Beijing, China

[4] China-Japan Union Hospital of Jilin University, Jilin, China

[*] These authors contributed equally to this work.

Corresponding authors
Wenqi Song,
songwenqi1218@163.com
Wei Sun, sunwei@ibms.pumc.edu.cn

## ABSTRACT

Previous studies reported that sex and age could influence urine metabolomics, which should be considered in biomarker discovery. As a consequence, for the baseline of urine metabolomics characteristics, it becomes critical to avoid confounding effects in clinical cohort studies. In this study, we provided a comprehensive lifespan characterization of urine metabolomics in a cohort of 348 healthy children and 315 adults, aged 1 to 78 years, using liquid chromatography coupled with high resolution mass spectrometry. Our results suggest that sex-dependent urine metabolites are much greater in adults than in children. The pantothenate and CoA biosynthesis and alanine metabolism pathways were enriched in early life. Androgen and estrogen metabolism showed high activity during adolescence and youth stages. Pyrimidine metabolism was enriched in the geriatric stage. Based on the above analysis, metabolomic characteristics of each age stage were provided. This work could help us understand the baseline of urine metabolism characteristics and contribute to further studies of clinical disease biomarker discovery.

## INTRODUCTION

In recent years, urine metabolomics has been widely used in disease biomarker discovery (*Thevenot et al., 2015*; *Chen & Kim, 2016*). Previous studies have reported that sex and age could influence urine metabolomics, which should be considered during biomarker discovery (*Reusch et al., 2018*). Therefore, understanding the baseline of urine metabolomics characteristics would be helpful for better understanding metabolism status under healthy conditions and discovering disease-specific metabolism disorders.

Investigation of urine metabolomics variation in a healthy population has been performed in children and adults using various approaches, including nuclear magnetic resonance (NMR) and mass spectrometry coupled to either gas (GC-MS) or liquid chromatography (LC-MS). In 2016, urinary metabolites from 30 healthy children were assessed at 6 months and 1, 2, 3, and 4 years of age using NMR spectroscopy. Amino acid metabolism was significantly different between infants aged 6 months and 1 year, whereas carbohydrate metabolism was significantly different between children aged 2 and 3 years (*Chiu et al., 2016*). In 2018, one research on metabolome differences in children aged 6 to 11 from six European centers was performed using NMR and targeted LC-MS/MS. Both urine and serum metabolomes were found to be associated with age, sex, BMI, and dietary habits (*Lau et al., 2018*). For the metabolic phenotypes in adults, in 2015, Etienne A. et al. characterized the urinary metabolomes of 183 healthy subjects using an LC-MS platform. A total of 108 metabolites related to amino acid metabolism, the carnitine shuttle, and the TCA cycle were found to be affected by age, sex, or body mass index (BMI) (*Thevenot et al., 2015*). Recently, in 2018, *Fan et al. (2018)* applied metabolomics profiling on urine samples from 60 healthy males and females using gas chromatography—time of flight mass spectrometry (GC-TOF-MS). Saturated fatty acids, tricarboxylic acid (TCA) cycle intermediates, and butyrate were found to be significantly related to the effect of sex (*Fan et al., 2018*). In 2018, a urine metabolomics study in 203 healthy adults was conducted by our team to find age- and sex-dependent urine metabolites. Metabolic pathways, such as tryptophan metabolism, citrate cycle, and pantothenate and CoA biosynthesis, were found to be related to sex and age (*Liu et al., 2018*). Despite the large efforts in characterizing urine metabolomics in healthy subjects, most of the previous studies were focused on the confounding factors of the urine metabolome. While the metabolic characterization of a population at each stage across the life span was unavailable until the present study.

In the present study, a large sample size including 348 children (aged from 1 to 18 years) and 315 adults (aged from 20 to 78 years) of balanced sex from China was enrolled. The goal is to characterize urine metabolic features in different age stages for males/boys and females/girls. High-resolution liquid chromatography-mass spectrometry (HRLC-MS)—based metabolic profiling was utilized to characterize the urine metabolome of each individual. In addition to sex- and age-related metabolites and pathways, the key metabolites that played important roles during different sex or age stages were identified and analyzed. Identification of these significant metabolites and metabolite networks would help understanding the physiological functions during different life stages. Additionally, the urine metabolomics atlas of normal healthy populations was characterized to serve as a reference database for future disease biomarker studies. It is suggested that the effects of sex or age-related metabolites should be considered for disease biomarker discovery. Metabolites with good stability in a healthy population are more suitable to serve as candidate disease biomarkers. Also, these metabolites associated with age and sex can be used as candidate biomarkers for growth, development, or aging studies.

**Table 1  Basic characteristics of the participants in this study.**

| Age stage | Female | Male | Total |
|-----------|--------|------|-------|
| 1–6 | 54 | 59 | 113 |
| 7–12 | 59 | 57 | 116 |
| 13–18 | 59 | 60 | 119 |
| 20–30 | 38 | 43 | 81 |
| 31–50 | 72 | 68 | 140 |
| >50 | 54 | 40 | 94 |
| Total | 336 | 327 | 663 |

## METHODS

### Study sample population

Our study enrolled 348 children aged 1 to 18 years from Beijing Children's Hospital and 315 adults aged 20–70 years from the 3rd Clinical Hospital, Jilin University (Table 1). According to physiological developmental characteristics, the enrolled population was divided into six age groups for males/boys and females/girls: preschool (aged 1–6), primary school (aged 7–12), secondary school (aged 13–18), youth (aged 20–30), middle (aged 31–50) and geriatric (aged > 50). Participants were checked and examined by trained nurses according to standard operating procedures. All physical examination indexes were in the normal range. On the day of the examinations, urine samples were collected at approximately 7:00 and 9:00 a.m. on an empty stomach. This study was approved by the Ethics Committee of Peking Union Medical College, and the project number is 047-2019. A doctor informed the eligible participants about the nature of the study and invited them to participate. All adults subjects provided oral informed consent, and informed consent was obtained from guardians of the children before participating in this study. There are no incentives or finances to participate. The research was carried out according to The Code of Ethics of the World Medical Association (Declaration of Helsinki). and the institutional review board of Peking Union Medical College has approved the study.

### Urine sample preparation

Urine samples were processed according to our previous protocols. Briefly, proteins from urine samples (200 µL) were precipitated using acetonitrile (400 µL). The mixture was vortexed for 30 s and centrifuged at 14,000 g for 10 min. The supernatant was dried under vacuum and then reconstituted with 200 µL 2% acetonitrile/water. Quality control (QC) samples were prepared by mixing aliquots of 50 representative samples across different groups (eight to nine samples were randomly selected across the six groups) of children and adults population to be analyzed. Samples from these two centers were analyzed one by one. Samples of the center were randomly injected into the LC-MS system. Before sample analysis, three QC injections were performed to equilibrate the LC system. And then QC sample was injected every ten or twelve samples to monitor system stability.

## Urine metabolite LC-MS measurements

HRLC-MS was selected for urinary metabolite detection due to its high sensitivity and reproducibility. Urine metabolite separation and analysis were conducted using a Waters ACQUITY H-class LC system coupled with an LTQ-Orbitrap Velos Pro mass spectrometer (Thermo Fisher Scientific, Waltham, MA, USA). The following 18-min gradient on a Waters HSS C18 column ($3.0 \times 100$ mm, $1.7$ μm) at a flow rate of $0.5$ mL/min was used: 0–1 min, 2% solvent B (mobile phase A: 0.1% formic acid in $H_2O$; mobile phase B: acetonitrile); 1–3 min, 2–55% solvent B; 3–8 min, 55–100% solvent B; 8–13 min, 100% solvent B; 13–13.1 min, 100–2% solvent B; 13.1–18 min, 2% solvent B. The column temperature was set at 45 °C.

All samples were untargeted scanned from 100 to 1,000 m/z at a resolution of 60 K at the positive ESI mode. The automatic gain control (AGC) target was $1 \times 10^6$, and the maximum injection time (IT) was 100 ms. The extracted MS features were divided into several targeted lists and imported to the MS2 method for targeted MS/MS analysis. MS/MS fragment acquisition was acquired at a resolution of 15 K with an AGC target of $5 \times 10^5$. The collision energy was optimized as 20, 40, or 60 for each targeted list with higher-energy collisional dissociation (HCD) fragmentation. The injection order of urine samples was randomized to reduce any experimental bias. The QC sample was injected regularly to monitor system stability.

## Statistical analysis

Raw data files were processed by Progenesis QI (Version 2.0, Waters, Milford, MA) software based on a previously published identification strategy, which included sample alignment, peak picking, peak grouping, deconvolution, normalization by total compounds, and final information export (*Zhang et al., 2016*). The exported data were further preprocessed by MetaboAnalyst 3.0 (http://www.metaboanalyst.ca), which included missing value estimation, log transformation, and Pareto scaling. Variables that were missed in 50% or greater of the samples were removed from further statistical analysis (*Luier & Loots, 2016*).

Nonparametric tests (Wilcoxon rank-sum tests and Kruskall–Wallis tests) were used to evaluate the significance of variables related to sex and age using MetaboAnalyst 3.0. Benjamini–Hochberg correction was applied throughout to account for multiple test comparisons. Cutoff of false discovery rates (FDR) 0.05 was applied (*Wilcoxon, 1946*). Pattern recognition analysis (principal component analysis (PCA) and orthogonal partial least square analysis (OPLS-DA)) was carried out using SIMCA 14.0 software (Umetrics, Sweden) to visualize group classification and select significant features (*Kalogiouri et al., 2020*). Permutation tests ($n = 100$) were used to validate the OPLS-DA and PLS-DA models to avoid over-fitting of the model. Significantly differential metabolites were chosen according to the following criteria: (i) adjusted p less than 0.05; (ii) fold change between two groups greater than 2; and (iii) the variable importance in the projection (VIP) value obtained from OPLS-DA was greater than 1. Heat map virtualization and metabolic pathway enrichment analysis were performed by MetaboAnalyst 3.0, and the enrichment results were visualized using R, based on previous methods (*Wang et al., 2018*).

## Feature annotation and metabolite identification

Differential features were divided into several lists and imported to "MS2 method" as including lists for targeted MS/MS analysis to identify the differential feature. The MS/MS spectra were further imported to Progenesis QI for metabolites annotation. In the present study, three databases were used for MS/MS matching: (1) Metlin MS/MS library (Waters, version 1.0.6499.51447, commercial, composed of authentic standards spectra obtained by orbitrap and TOF mass spectrometry). (2) In-house standards library (composed of ∼600 authentic standards spectra obtained by Thermofisher LTQ orbitrap and AB sciex QTOF5600 mass spectrometry in our lab; this library was constructed using Progenesis QI software). (3) Fragment library constructed using theoretical fragments calculated by a theoretical fragmentation algorithm: the "MetFrag" algorithm (*Allen et al., 2014*). Detailed compound identification information (.csv file) included compound ID, adducts, formula, score, MS/MS score, mass error (in ppm), isotope similarity, theoretical isotope distribution, web link, and m/z values. Confirmation of differential compounds was performed by the parameter of score value, calculated using the mean of five similarity metrics (each metric account for 20%), incluidng mass error, isotope similarity, and fragmentation similarity, retention time similarity, CCS Similarity. Herein, the search method does not support searching by retention time and CCS, thus the two metrics are 0. Therefore, the maximum value of the score is "60". The score value ranged from 0 to 60. According to the score results of the reference standards, the threshold of score value was set at 35.0, and compound isotope similarity is greater than 80%. The compound identification is more reliable with higher score values obtained. Metabolites annotation results were shown in Table S1A and Fig. S3. Metabolites function annotation was performed using KEGG metabolism pathway database, combined with manual annotation by reference searching.

# RESULTS

## Data quality control

The workflow for the present study is shown in Fig. 1. To reduce experimental variations from the sampling process, standard sampling procedures, including sampling time and sampling processing, were performed by a specialist. Data from the two centers' samples were analyzed separately. Overall, 730 injections were carried out (663 sample injecions and 67 QC injections). PCA was performed to evaluate the variation of samples (Fig. S1A). A good cluster of QC samples indicated good stability of the analytical platform.

## Sex-dependent metabolomics in children and adults

Children and adult urine samples were collected from two clinical centers, and sex-dependent metabolites were assessed in the children and adult samples, respectively. A total of 274 metabolites were identified and subjected to further analysis (Table S1).

Urine metabolic differences between males/boys and females/girls were assessed using PCA and OPLS-DA models. Scatter plots showed that sex differences in the urine metabolites of adults are more apparent than in children (OPLS-DA: children: R2Y = 0.599, Q2 = 0.379 ; adults: R2Y = 0.928, Q2 = 0.768. Figs. S1B and S1C). Metabolites with
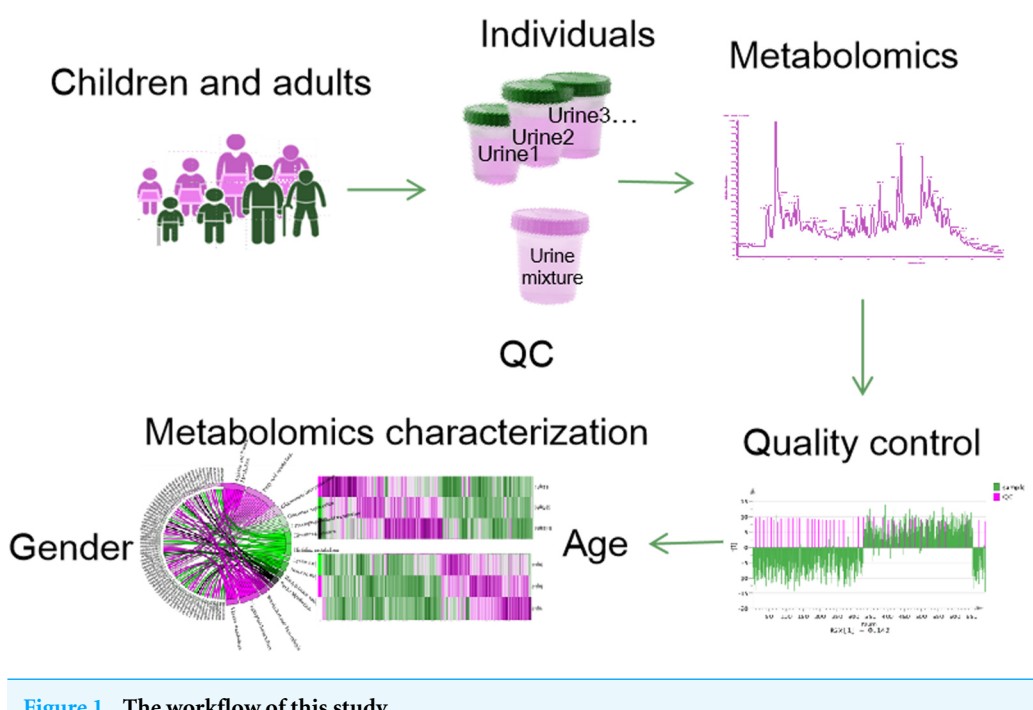

**Figure 1** The workflow of this study.

VIP values above the threshold value 1 and $p$ values below the significance threshold were considered sex-dependent. A total of 42 metabolites were found to be significantly different between boys and girls, and 98 metabolites were different between males and females (Figs. 2A and 2B, Tables S2 and S3). A total of 17.9% (21) of these metabolites showed sex differences in both children and adult populations with the same change trend (Fig. 2C). Guanidoacetic acid, 5-hydroxyindoleacetic acid, dopamine, 5′-methylthioadenosine and indoleacrylic acid showed higher levels in females/girls. The metabolites deoxyinosine, cotinine glucuronide, dopamine glucuronide and L-formylkynurenine showed higher levels in males/boys. Sex differences of these common differential metabolites in the adult population were larger than those in the children population (Fig. 2C). Specifically, the metabolites cortisol, uric acid, 18-hydroxycortisol, deoxycholic acid and glycine conjugate were found to be sex-dependent only in the children population. Metabolites of deoxyuridine, pantothenic acid, riboflavin, and 3-hydroxytetradecanedioic acid were sex-dependent only in the adult population. These metabolites suggested differential metabolic status between children and adults.

Furthermore, sex differences during each age stage were examined. The $p$ values of OPLS-DA for sex separation during each stage were used to evaluate the significance of sex differences. The results showed a parabolic trend of sex differences during life span, with less significance in the pre- and primary school stages, high significance during the secondary school, youth and middle stages, and less significance during the geriatric stage (Fig. 2D).

Pathway enrichment analysis would provide an overview of sex-dependent metabolism status in children and adults. Arginine, proline, tryptophan metabolism and glucuronate

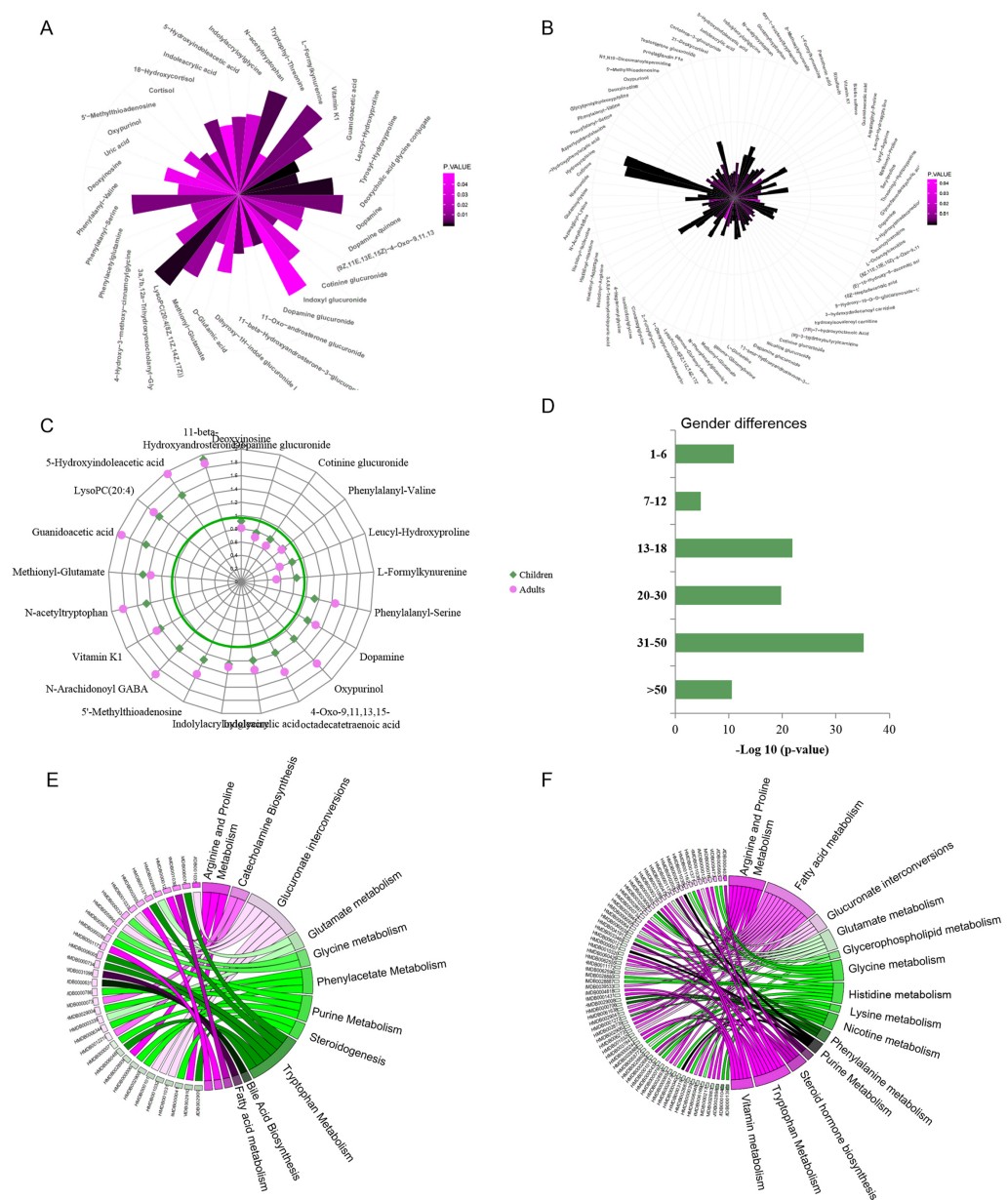

**Figure 2** (A–F) Representative gender-differential metabolites and metabolism pathways in children and adults.

interconversions were found to be sex-dependent in both the children and adult populations (Figs. 2E and 2F). Additionally, several specific pathways were found to be sex-dependent in adults only, including nicotine metabolism, nicotinate and nicotinamide metabolism, steroid hormone biosynthesis, lysine metabolism and histidine metabolism. These results indicated the commonness and differences of sex-dependent metabolism characteristics in children and adults, probably resulting from different physiological characteristics, dietary

habits or occupational hazards (*Biswas et al., 2021*). The detailed sex-associated pathways in each age stage are listed in Table S4.

## Age-dependent metabolomics in children and adults

Age is another important factor in metabolism status. PCA was firstly performed, showing apparent separation of different groups (Fig. S2A). To determine metabolites related to age, PLS-DA modeling was performed on urine metabolite profiles of populations with different age stages. Since the metabolism status of the population was different between sexes, the analyses for age in children and adults were performed separately for males/boys and females/girls.

In the children population, the PLS-DA three-component score plot showed clear associations of metabolite profiles with age in boys (R2Y = 0.719, Q2 = 0.528, Fig. 3A, Fig. S2B), indicating significant metabolism differences among different age stages. The same trend was observed in girls (R2Y = 0.78, Q2 = 0.564, Fig. 3B, Fig. S2B). In both boys and girls, two clusters of metabolites were found, one cluster with metabolites decreasing with age (Component 1); the second cluster with metabolites showing the highest level in the primary school aging 7 to 12 (Component 2) (Fig. 3C). The first cluster contributes most to age variation in both girls and boys. In the adult population, clear separations were also observed for the three age groups in males and females (PLS-DA-male: R2Y = 0.652, Q2 = 0.17; PLS-DA-female: R2Y = 0.583, Q2 = 0.362, Figs. S2B–S2D). Similar to children, metabolites decreasing with age contributed most to age differences (Fig. S2E). According to the significance threshold, a total of 98 and 76 metabolites were selected as age-dependent in boys and girls, respectively. For the adult population, 55 and 80 metabolites were age-dependent in males and females, respectively (Tables S5–S8). These metabolites were submitted for further pathway analysis.

In the children population, the metabolites involved in histidine metabolism, riboflavin metabolism, pantothenate and CoA biosynthesis and fatty acid biosynthesis were found to change with age in both boys and girls. The involved metabolites, including N-acetylhistidine and pantothenate decrease with age, while riboflavin and dodecanoic acid increase with age. Tryptophan metabolism was age-dependent in boys. In the adult population, most differential pathways were the same as in the children population. However, in adult males, steroid hormone biosynthesis was found to be age-dependent, although this difference was found in girls rather than boys. The detailed age-dependent metabolism pathways for males/boys and females/girls are listed in Table S4.

Metabolites showing the highest level for each group indicate the specific metabolism characteristics during a certain life stage. For each age stage, representative pathways were listed for males/boys and females/girls (Table 2). In the population aged 1–6, metabolites related to energy metabolism (pantothenate and CoA biosynthesis, beta-alanine metabolism) showed a high expression level. In the population aged 7–12, metabolites involved in lipid, glucose and amino acid metabolisms showed higher levels. In the population aged 13–18, the metabolic features in boys and girls showed some differences. Spermidine and spermine biosynthesis and riboflavin metabolism showed high activity in boys. In addition, fatty acid oxidation and biosynthesis were active in girls.

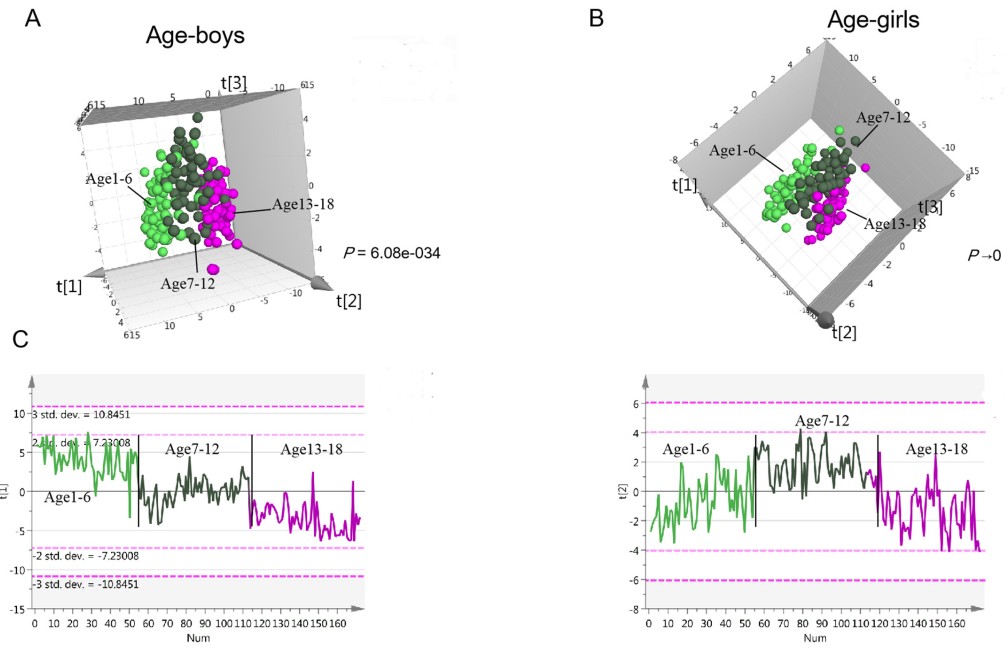

**Figure 3  Urine metabolomics variation with age in children.** (A) PLS-DA score plot of urine metabolomics of different ages in boys. (B) PLS-DA score plot of urine metabolomics of different ages in girls. (C) Change trends of the first and second principal components of PLS-DA in children; these two principal components contribute most to the age separation in children.

In populations aged 20–30, organ metabolism was the most active. Androgen and estrogen metabolism, steroidogenesis and arachidonic acid/linoleic acid metabolism showed higher activity. During the 30- to the 50-year-old stage, males and females showed different metabolic features. In males, vitamin B6 and purine metabolism were active. In females, steroidogenesis and caffeine metabolism were the main metabolic features. For the last age group, males and females showed similar metabolic features, with high levels of the metabolites vitamin K, pyrimidine and caffeine.

## DISCUSSION

In the present study, we tested urine samples from a large population of children and adults to characterize metabolic features in different age stages. Urine metabolism characteristics from early life to geriatric age were profiled, which is the first systematic comparative study based on such a large and healthy population. Urine volume can vary widely based upon water consumption and other physiological factors. As a result, the concentrations of endogenous metabolites in urine vary widely and normalizing for these effects is necessary. Reliable identification of features distinguishing biological groups in urinary metabolite fingerprints requires the control of total metabolite abundance. Normalizations to creatinine values, osmolality, and total compound (useful MS signals) were the commonly used normalization techniques for overcoming sample variability in urinary metabolomics. Previous researches have compared the effect of different normalization method on urine

**Table 2  Metabolic characteristics in males (boys) and females (girls).**

| Age ranges | Males (boys) | Females (girls) |
|---|---|---|
| 1–6 | Pantothenate and CoA biosynthesis | Pantothenate and CoA biosynthesis |
| | Alanine metabolism | Alanine metabolism |
| | Pyrimidine metabolism | Vitamin B6 metabolism |
| 7–12 | Tryptophan metabolism | Ether lipid metabolism |
| | | Nicotinate and nicotinamide metabolism |
| | | Histidine metabolism |
| 13–18 | Fatty Acid oxidation and biosynthesis | Fatty acid oxidation and biosynthesis |
| | Spermidine and spermine biosynthesis | |
| | Riboflavin metabolism | |
| 20–30 | Androgen and estrogen metabolism | Androgen and estrogen metabolism |
| | Steroidogenesis | Steroidogenesis |
| | Linoleic acid metabolism | Arachidonic acid metabolism |
| 30–50 | Vitamin B6 metabolism | Steroidogenesis |
| | Purine metabolism | Caffeine metabolism |
| >50 | Vitamin K metabolism | Vitamin K metabolism |
| | Steroidogenesis | Pyrimidine metabolism |
| | Caffeine metabolism | |

metabolomics study. It showed that no normalization and normalization to creatinine yielded a highly similar number of significant features measured by LC-MS, whereas normalization to osmolality yielded more significant features. The best performance was observed with normalization to the total compound. The latter approach does not only correct for different urinary outputs, but also accounts for injection variability (*Warrack et al., 2009*; *Vogl et al., 2016*). Thus, in the present study, we normalized the data using the method of "normalizing to the total compound".

## Sex-dependent metabolism status in children and adults

In the present study, urine metabolomes of males/boys and females/girls in children and adults were characterized. Consistent with previous studies (*Chiu et al., 2016*; *Fan et al., 2018*; *Liu et al., 2018*), amino acid metabolism, including tryptophan metabolism and arginine and proline metabolism, showed sex differences in both children and adults. In females/girls, tryptophan metabolites, indoleacrylic acid, indolylacryloylglycine and 5-hydroxyindoleacetic acid showed higher levels compared with males/boys. Metabolites of 5-hydroxyindoleacetic acid showed higher levels in females/girls than males/boys across the life span (Fig. 4A), indicating high activity of tryptophan metabolism in females/girls. 5-hydroxyindoleacetic acid is one product of tryptophan metabolism. The ability of tryptophan availability showed sex differential, which was mediated by steroids and immune activation. It was reported that the effects of immune activation on tryptophan degradation are significantly more pronounced in females than males, and therefore, females may show higher levels of tryptophan products (*Akhmadeev & Kalimullina, 2013*; *Songtachalert et al., 2018*). Metabolites of phenylalanyl-valine showed higher levels in males/boys than in females/girls. Phenylalanyl-valine is a dipeptide probably from protein
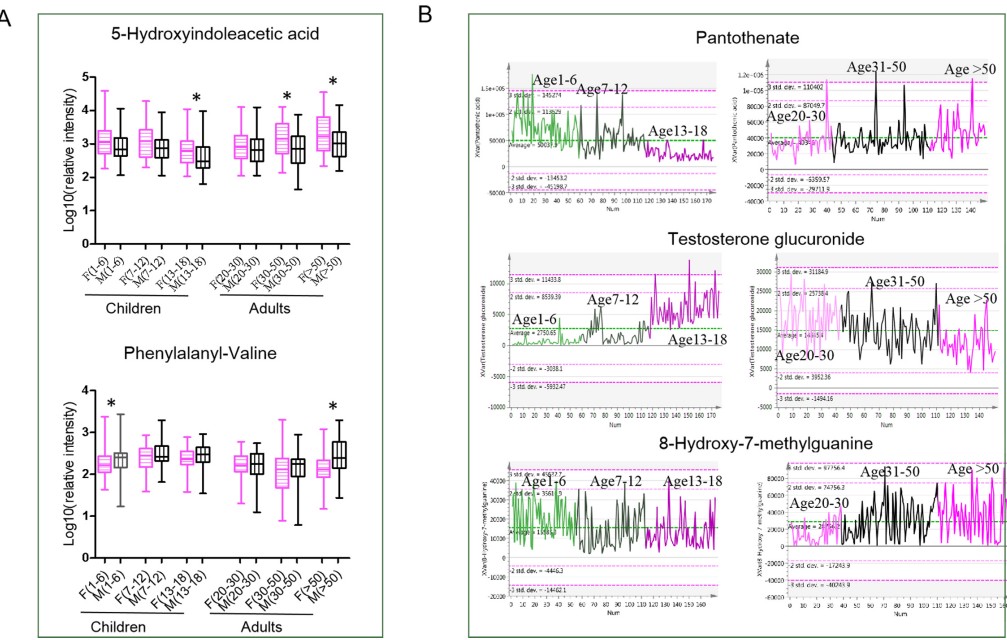

**Figure 4** (A–B) Metabolic characteristics for different gender and age stages.

synthesis and degradation process. Some dipeptides are known to have physiological or cell-signalling effects although most are simply short-lived intermediates on their way to specific amino acid degradation pathways following further proteolysis (*Mizushige, Uchida & Ohinata, 2020*). Up to date, there are no reports showing physiological or cell-signalling effects of phenylalanyl-valine. Thus, we speculate the higher level of phenylalanyl-valine in males/boys perhaps just the consequence of proteolysis process differences between males/boys and females/girls.

In addition, children and adults showed different sex-dependent metabolic statuses. In the child population, purine metabolites, steroid metabolites and amino acid metabolites showed specific sex dependence. For example, uric acid, an oxidation product of purine, showed a significantly higher level in boys compared with girls but showed no sex variations in adults. Renal excretion of uric acid in children differs from that in adults. It was reported that the younger the child, the greater the excretion of uric acid (*Baldree & Stapleton, 1990*). Altered urine uric acid level is an indispensable marker in detecting rare inborn errors of metabolism (*Jasinge et al., 2017*). The level of uric acid in urine was higher in boys than girls, probably contributing to the higher prevalence of hyperuricemia in boys than in girls (*Niegawa et al., 2017*). Uric acid is a risk factor of hyperuricemia. Increased uric acid leads to precipitation of monosodium uric acid crystals, which can cause uric acid kidney stones and gout. The overall kidney function seems to be improved in females when compared to males, contributing to rapid metabolism of uric acid in females (*Halperin Kuhns & Woodward, 2020*). In addition, a clear sex difference has emerged in the physiological regulation of urate homeostasis. Specifically, estrogen may play a role in the regulation of expression or activity of uric acid transporters, ABCG2 and SLC2A9. Reports demonstrate

that SLC2A9 had a stronger association with lower uric acid levels in females, whereas ABCG2 had a stronger association with higher uric acid levels in males, demonstrating that regulation and/or activity of these transporters may be influenced by factors that contribute to biological sex (*Dehghan et al., 2008*).

Compared to children, more specific sex-dependent metabolites in adults were identified. Most of the specific sex-dependent metabolites in adults showed higher levels in males, including fatty acids, acylcarnitines, steroid hormones and dipeptides. Higher levels of acylcarnitines in males were found in our previous study, indicating a higher activity of fatty acid oxidation and energy production in males compared to females (*Liu et al., 2018*). In addition, a higher level of steroid hormone biosynthesis was found in males, consistent with previous studies (*White et al., 1991*).

## Age-dependent metabolism status in children and adults

For children and adults, many common age-dependent pathways between boys/males and girls/females were discovered, indicating common metabolism status variations with age in humans. For instance, pantothenate and CoA biosynthesis, fatty acid biosynthesis and tryptophan metabolism were found to change with age in both children and adults. These pathways correspond to energy demand changes with aging (*Chiu et al., 2016*). As shown in Fig. 4B, in the children population, pantothenate decreases with age, showing the highest level in children aged 1–6, and in adults, pantothenate increases with age. Pantothenate is a vitamin required to sustain life. Pantothenate is needed to form CoA and is thus critical in the metabolism and synthesis of carbohydrates, proteins, and fats. A previous study showed that pantothenic acid is a lifespan-extending agent. It may have an anti-aging effect by itself or by synergizing the action of other vitamins (*Kunugi & Ali, 2019*). However, the exact mechanism through which pantothenate may extend lifespan is not well-understood. Interestingly, tryptophan metabolism was only found to be age-dependent in boys and males, while no significant changes were found in girls and females. Metabolites of tryptophanol and 5-methoxytryptophan were reported to be associated with increased cellular anti-inflammatory and blood circulation properties (*Cheng et al., 2012*), probably reflecting age-associated metabolism activity differences between sexes. In addition, it found that the pathway of steroid hormone biosynthesis was changed with age in adults and children, although this difference was found in girls rather than boys, partly resulting from later sexual development in boys. Steroids are characteristic metabolites and showed the highest level in the youth life stage. Testosterone glucuronide is the downstream metabolites of testosterone. The level of testosterone glucuronide was very low in children and increased with puberty and youth. In the adult population, a high level of testosterone glucuronide was also observed in the youth and then decreased with aging in the geriatric (Fig. 4B). Steroid level changes were tightly associated with sexual development during the life span (*Steensma et al., 2013*).

Additionally, children and adults showed different age-dependent metabolic statuses. Particularly in the children population, the fatty acid biosynthesis pathway was age-dependent. The results showed a positive correlation with increasing age. Fatty acids constitute a large energy source for the body. Increased fatty acid metabolism indicated

high ATP generation with age in a children population (*Wakil & Abu-Elheiga, 2009*). Fatty acid, dodecanoic acid, modulates gastrointestinal functions, including gut hormones and pyloric pressures, which are important for the regulation of energy intake, and both potently suppress energy intake (*McVeay et al., 2019*). In adults, pyrimidine metabolism and caffeine metabolism were found to be age-dependent. Pyrimidine metabolism was found to be positively correlated with aging, showing the highest level in geriatric adults. Deoxyuridine, a naturally occurring nucleoside, is considered an antimetabolite that is converted to deoxyuridine triphosphate during DNA synthesis. Disturbance of DNA synthesis may modulate the aging process and contribute to the high incidence of cancer with aging (*Kirsh, Cutler & Hartman, 1986*).

## Metabolomic characteristics during each age stage
### Metabolomic characteristics during the pre- and primary school stages
During early life in the preschool and primary school stages, sex differences were relatively smaller than in other age stages, as shown in Fig. 4. During the preschool stage, pathways of pantothenate and CoA biosynthesis, pyrimidine metabolism, vitamin B6 and alanine metabolism showed high activity in girls and boys. These active pathways were associated with energy and nutrient supply. These metabolic characteristics correspond to the physiological characteristics during this life stage, high metabolism activity for rapid growth and development demands (*Chiu et al., 2016*). Research on children aged 2–7 years old suggested that pantothenate and CoA biosynthesis, pyrimidine metabolism and vitamin B6 metabolism were significantly related to autism spectrum disorder (ASD) (*Gevi et al., 2016*). These results highlight the importance of these pathways to maintain homeostasis during preschool.

During the primary school stage, the most active metabolism pathways are tryptophan metabolism, lipid metabolism, nicotinate and nicotinamide metabolism and histidine metabolism. These metabolic features were corresponding to the main physiological characteristic-visual development, blood circulation increasing and rapid metabolism activity during this stage. Although sex differences were small during the primary school stage, metabolites with the highest level in boys and girls showed specific features. In boys, tryptophan metabolites, such as 5-methoxytryptophan, were found to show the highest level during primary school stage compared with other age groups. 5-Methoxytryptophan is an endogenous tryptophan metabolite with anti-inflammatory properties (*Cheng et al., 2012*). In addition, 5-methoxytryptophan was reported to be involved in the cyclic metabolism of the retina (*Leino & Airaksinen, 1985*), ventricular remodeling and maintaining liver function (*Rossignol & Frye, 2011*; *Lin et al., 2016*). In girls, urocanic acid, a breakdown (deamination) product of histidine, showed the highest level. Urocanic acid is one of the essential components of human skin (*Wezynfeld et al., 2014*). It could accumulate in the epidermis and maybe both a UV protectant and an immunoregulator. A higher level of these metabolites in girls may contribute to skin development during this age stage. It was reported that skin disorders are more common among girls than boys aged 6 to 17 years (*Sula et al., 2014*), which could be affected by the immunomodulatory effects of urocanic acid (*Finlay-Jones & Hart, 1997*).

### Metabolomic characteristics during adolescence and youth

During adolescence and the youth stage, sex differences become more significant, partly due to changes in hormone and endocrine levels. The main metabolic feature during these stages was fatty acid oxidation and biosynthesis, androgen and estrogen metabolism and steroidogenesis. These metabolic characteristics correspond to pubertal development, neurodevelopmental changes and heightened stress sensitivity during adolescence and youth stages. Cortisol, androstenol, testosterone, and their glucuronide metabolites showed higher levels during this period. Cortisol is the main glucocorticoid secreted by the adrenal cortex and is involved in the stress response. Synergies between cortisol reactivity and testosterone were reported to influence antisocial behavior in young adolescence. The youth with high diurnal testosterone, combined with high or moderate cortisol reactivity, were significantly higher on antisocial behavior and attention behavior problems (*Susman et al., 2017*).

In addition to the common metabolic features during adolescence and the youth stage, boys and girls also showed specific age-associated metabolic characteristics. During adolescence, spermidine biosynthesis was higher in boys. One of the involved metabolites was 5-methylthioadenosine, a byproduct of polyamine synthesis in DNA turnover cycles that increases with inflammation to modulate cellular stress. It has been shown to influence the regulation of gene expression, proliferation, differentiation, and apoptosis (*Avila et al., 2004*). Higher serum levels of 5-methylthioadenosine have been reported in youth boys when compared to girls (*Perng et al., 2017*). A direct association between 5-methylthioadenosine and high metabolic risk was found in boys, possibly driven by proinflammatory pathways (*Guasch-Ferre et al., 2016*). While in adolescent girls, fatty acid oxidation and biosynthesis showed high activity. Acylcarnitines showed the highest level during the adolescent stage compared to other age groups, indicating high activity of carnitine acetyltransferase in mitochondria (*Zammit et al., 2009*). These results indicated the preferred metabolic fuel from fatty acid oxidation in adolescent girls (*Zammit et al., 2009*).

In the youth males aged 20–30 years, linoleic acid metabolites showed higher levels in addition to steroid metabolism. The involved metabolite eicosapentaenoic acid serves as the precursor for prostaglandin-3. It could enhance the production of the cytoprotective prostanoid 15-deoxy- prostaglandin J2 (15d-PGJ2) (*Davidson, Higgs & Rotondo, 2013*), which corresponds to the elevated prostaglandin production in youth males (*Pace et al., 2017*). Arachidonic acid metabolites, 14,15-DiHETrE, showed the highest level during youth age in females. 14,15-DiHETrE is a cytochrome P450 eicosanoid. P450 eicosanoids are involved in the regulation of vascular tone, renal tubular transport, cardiac contractility, cellular proliferation, and inflammation. Regulation of P450 eicosanoid levels is determined by the induction or repression of the cytochrome P450 enzymes responsible for their formation (*Ng et al., 2007*). According to a previous study, an approximate 3.5% decline in CYP450 content occurs for each decade of life (*Trenaman et al., 2021*). Therefore, a higher 14,15-DiHETrE level in youth age probably results from higher CYP 450 content at the young.

### Metabolomic characteristics during the middle and geriatric stages

During the middle age stage, males and females showed the most significant sex differences. In males, the main metabolic features were vitamin B6 and purine metabolism. The level of 8-hydroxy-7-methylguanine remained fairly constant from early life to youth. However, it increased significantly during the midlife stage and reached the highest level during the geriatric stage (Fig. 4B). Hydroxy-7-methylguanine is an endogenous methylated nucleoside. Methylated nucleoside in urine could be derived from covalently bound adducts in DNA, which are formed by exposure to various carcinogens. High methylated purine bases were found in tumor-bearing patients compared to healthy controls (*Morris, Simmonds & Davies, 1986*). Urine alkylated purines were partly derived from covalently bound adducts in DNA formed from exposure to carcinogenic alkylating agents (*Choi & Guengerich, 2006*). Purine disorders may be associated with serious, sometimes life-threatening consequences. Based on above reports, it is predicted that a higher level of hydroxy-7-methylguanine in geriatric stages probably results from the accumulation of lots of carcinogenic substances along with age. In females, the metabolism pathway of steroidogenesis and caffeine metabolism showed high activity. Menopausal symptoms are an unavoidable problem in females during this period, which could contribute to some metabolic disorders. Caffeine metabolism was reported to be associated with menopausal symptoms, particularly vasomotor symptoms (*Faubion et al., 2015*).

For the above 50 years group, the sex difference decreased. During this period, organ metabolism activity gradually slows down. Energy-supply metabolism pathways, such as fatty acids and amino acids, showed low levels. In contrast, steroidogenesis, caffeine and pyrimidine metabolism showed high levels. Higher levels of pyrimidine metabolites in the old stage have adverse effects on health (*Choi & Guengerich, 2006*). Additionally, several cognitive impairment-related metabolites, including acetylhistidine and steroid hormones, were found to be higher in the old population, partly contributing to the high incidence of cognitive impairment at older ages (*Bressler et al., 2017*).

## CONCLUSION

In conclusion, a comprehensive view of human metabolism status across the life span was provided in the present study. To our knowledge, this is the first systematic study to analyze metabolism characterization based on a population across a considerable age range. This study showed that sex differences existed from early life stages, and these differences were much smaller than those in adults. Age is another recognized confounder. Metabolism characteristics for each age group could reflect the metabolism status during different life stages and possibly contribute to some age-dependent disease incidences. Our present study would be helpful to understand the age- and sex-dependent metabolism differences, which will be a critical component for the development of metabolomics-based systems biology as a population screening and precision medicine research. The raw data is available at ProteomeXchange: PXD034390.

Additionally, several limitations still exist and need to be further validated in the future. First, urine samples of children and adults were collected from two different hospitals.

Although sampling operation is strictly controlled, multi-center effects is unavoidable. Thus, data from two centers were analyzed individually. Second, in the children population, each stage could show specific features for their rapid development. However, due to sample size limitations, we provided an overview of the average metabolism status over a period of 6 years. Third, the influences of diets and circadian rhythm on urine metabolomics could not be completely eliminated, though all subjects were from the same region. For future validation analysis, these influences should be systematically evaluated and analyzed.

### Funding
This work was funded by the National Natural Science Foundation of China (No. 82170524, 31901039), the Beijing Medical Research (No. 2018-7), the CAMS Innovation Fund for Medical Sciences (2021-1-I2M-016), the Special Fund of the Pediatric Medical Coordinated Development Center of Beijing Hospitals Authority (XTCX201815), the Beihang University and Capital Medical University Advanced Innovation Centre for Big Data-Based Precision Medicine Plan (BHME-201910) and Beijing Talents Fund (2018000021469G278). The funders had no role in study design, data collection and analysis, decision to publish, or preparation of the manuscript.

### Grant Disclosures
The following grant information was disclosed by the authors:
National Natural Science Foundation of China: 82170524, 31901039.
Beijing Medical Research: 2018-7.
CAMS Innovation Fund for Medical Sciences: 2021-1-I2M-016.
Special Fund of the Pediatric Medical Coordinated Development Center of Beijing Hospitals Authority: XTCX201815.
Beihang University and Capital Medical University Advanced Innovation Centre for Big Data-Based Precision Medicine Plan: BHME-201910.
Beijing Talents Fund: 2018000021469G278.

### Competing Interests
The authors declare there are no competing interests.

### Author Contributions
- Xiaoyan Liu conceived and designed the experiments, prepared figures and/or tables, authored or reviewed drafts of the article, and approved the final draft.
- Xiaoyi Tian conceived and designed the experiments, authored or reviewed drafts of the article, and approved the final draft.
- Shi Qinghong performed the experiments, prepared figures and/or tables, and approved the final draft.
- Haidan Sun analyzed the data, prepared figures and/or tables, and approved the final draft.

- Li Jing performed the experiments, authored or reviewed drafts of the article, and approved the final draft.
- Xiaoyue Tang performed the experiments, prepared figures and/or tables, and approved the final draft.
- Zhengguang Guo analyzed the data, prepared figures and/or tables, and approved the final draft.
- Ying Liu performed the experiments, analyzed the data, prepared figures and/or tables, and approved the final draft.
- Yan Wang analyzed the data, prepared figures and/or tables, and approved the final draft.
- Jie Ma performed the experiments, analyzed the data, prepared figures and/or tables, and approved the final draft.
- Ren Na analyzed the data, prepared figures and/or tables, and approved the final draft.
- Chengyan He analyzed the data, authored or reviewed drafts of the article, and approved the final draft.
- Wenqi Song conceived and designed the experiments, authored or reviewed drafts of the article, and approved the final draft.
- Wei Sun conceived and designed the experiments, authored or reviewed drafts of the article, and approved the final draft.

## Human Ethics

The following information was supplied relating to ethical approvals (*i.e.*, approving body and any reference numbers):

Institutional review board of Peking Union Medical College approved the study (047-2019).

## Data Availability

The raw data is available at ProteomeXchange: PXD034390.

## Supplemental Information

Supplemental information for this article can be found online at http://dx.doi.org/10.7717/peerj.13545#supplemental-information.

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
