# Peer review of "Characterization of LC-MS based urine metabolomics in healthy children and adults"

_PeerJ, doi:10.7717/peerj.13545_

## Round 0.1 · original submission · Major Revisions

The manuscript was reviewed by two expert reviewers who, with the exception of some minor as well as major suggestions, indicated that it has the potential to be accepted for publication. Can you please address all of these comments, and take special note of those more substantial suggestions of the second reviewer.

Reviewer 1 ·

Basic reporting

Raw data not shared, nor a link provided in the manuscript

General:
• It would be more "correct" to use the term "sex" instead of "gender" since the term "gender" (esp. in today's culture) is used more broadly to denote a range of identities that do not necessarily correspond to the established ideas of male and female.
• The authors consider consistency in terms of word-use throughout the manuscript, since they use the word “sex” in line 51+103, but “gender” elsewhere. This should be reviewed for other wording choices as well: “primary school” (lines 81, 210, 235, etc.) vs “second school” (line 449)
• “geriatric” is a better word choice than “old”.
• Define at first mention (blue highlight): BMI (line 51), TCA (line 54), TOF (line 57), HRLC (line 69), FDR (line 130), MSI (line 142); 15d-PGJ2 (line 408)
• Avoid personal pronouns/terms like “we” (line 58).
• Please see the PDF edits for some minor grammar/linguistic corrections: lines 23–25, 29, 47, 81, 162, 189, 237, 251, 278, 279, 283–284, 329, 340, 346, 359, 363, 368, 403, 406, 421, 425, 435, 451.

Abstract:
• The authors state “…gender-dependent urine metabolites are much greater in adults than in children” (lines 25-26), but this is in contrast with their summary which states “…urine metabolites showed larger gender differences in children than in adults” (no line numbers available, page 4)
• Please confirm the age of the adult cohort. In the summary the authors state “…aged 20-78”, but in the abstract (line 24, and also line 67 in the introduction) it is “…70”.

Introduction:
• The authors indicate stated what knowledge gap in the field they aim to fill, but should elaborate HOW this would be useful for, for example, future disease biomarker discovery? (line 74)

Methods:
• S2.4: The need for missing value replacement, transformation, scaling and a 50% filter needs to be referenced (lines 125-126). The authors might consider the work of Luies and Loots (DOI:10.1007/s11306-016-0969-x) who describes all of these approaches.

Experimental design

Methods:
• S2.1: How were the children informed on the study, in which case parents/guardians need to approve participation?
• S2.1: Were there any incentives to participate, financial perhaps?

• S2.2: How were the 50 samples used to compile the QC selected? The authors state that these were representative across the groups (line 98), does this mean that 8-9 samples were RANDOMLY selected across the six groups (mentioned in lines 81-82)?
• S2.2: Analysis was done over 12 days, how many batches were involved? How were these batches comprised, e.g. did they represent the entire cohort of various ages?
• S2.2: Did they analysis include ‘leading QCs’ (e.g. 3-5 injection prior to the first samples daily to ‘warm up’ the LC and thereby avoiding potential batch effects?

• S2.3: Why was positive ESI mode selected? Did the authors also consider negative ESI mode?

• S2.4: Please state the software version used (line 121).
• S2.4: The need for missing value replacement, transformation, scaling and a 50% filter needs to be referenced (lines 125-126). The authors might consider the work of Luies and Loots (DOI:10.1007/s11306-016-0969-x) who describes all of these approaches.
• S2.4: Referencing is needed for lines 127-133 that actually describes these statistical approaches, should the reader be interested.
• S2.4: The sentence starting in Line 133 (“100 permutation tests were used to…”) needs to be rephrased as sentences should not start with a number. Consider “Permutation tests (n=100) were used to…”
• S2.4: Please confirm that the symbol in line 136 should not be greater than OR EQUAL TO 2, instead (≥)
• S2.4: The description of “VIP” (line 136) needs to be corrected to “variable importance in the projection” (this does not stand for variable importance plot).

• S2.5: “…and imported to MS method for targeted data dependent analysis” (line 142) is confusing and needs to be rephrased. Is there perhaps a missing word(s) here?
• S5.2: What percentage isotope similarity (line 146/147) was considered sufficient?

• General: The conclusion makes reference to batch effects being an issue (lines 446–447), how was this corrected in the current dataset? There is no mention of this in the methods section.

Validity of the findings

Results:
• S3.1: How dit 663 samples result in 730 injections; were 67 QCs injected?
• S3.1: Principle component analysis was abbreviated as PCA in line 131. This abbreviation should be used in line 162.

• S3.2: The first 2 sentences of this section should be switched (lines 167-169).
• S3.2: Was the PLSDA not used as well?
• S3.2: The terminology (males vs boys; females vs girls) can cause some confusion. For example, do the authors refer to females as an entirety in line 179? In such case, they should clarify and denote this as "female/girls". If not, the terminology is correct as is. Also consider the same question for “males” in line 181.
• S3.2: Can the authors elaborate on what other type of environmental factors they refer to in line 196?

Major comments:
The data interpretation is scattered between Sections 3 and 4, making this somewhat repetitive, with a lot of ‘jumping between sex vs gender trends’. The Results section should simply list the results only (what the data showed based on what stats they did), without ANY explanations provided for any of the metabolic changes. This would be Sections 3.1–3.3 only, which still need to be re-evaluated/adapted to remove any explanations (e.g. lines 194–197; 226–227). In Sections 3.4–3.5 the authors started to explain some of their findings (e.g. lines 238–255). This should all form part of the Discussion section; hence these type of discussions need to be moved to/combined with the Discussion section. Consider the following feedback for all of these sections in light of such a restructure:

• S3.4: The first 2 sentences (lines 231–234) are redundant and repetitive. It should be removed and/or only mentioned in the Results section.
• S3.4: (line 242) The authors can exclude either Fig 4 or Table 2 from the manuscript, since these both give the exact same information — the reader would not miss out on any NEW information in the absence of one of these. Since Table 2 is easier to interpret for a reader who doesn’t understand heat maps, I would recommend removing Fig 4. (This Pathway Analysis part can remain part of the Results section, leaving the rest of this interpretation/discussion for Section 4) — Combine S3.4 interpretations with the discussion; combine the results parts with Section 3.

• Section 3.5 is somewhat repetitive of Sections 3.3 (last paragraph) and 3.4; this section should rather be combined with the rest of the Discussion, to expand on what is stated there. This way, the authors will not need to repeat the mentions of previous sections before providing and/or discussing the key variation further.
• S3.5: Why would this trend in lines 268-272 occur?
• S3.5: Did pantothenate/VitB5 (lines 273-276) decrease with age? Do we then need less B5 as we age?
• S3.5: Why would this trend in lines 283-287 occur?

• General: The authors should consider each of their differentiating metabolites/pathways (for sex and age) separately and interpret these on an individual level. For example, “…steroid hormone biosynthesis was found to be age dependent, although this difference was found in girls rather than boys, partly resulting from later sexual development in boys” (lines 225-227). Here, the authors give an (although very brief/superficial) EXPLANATION of the increased steroid hormone biosynthesis observed. There is no in-depth or even a generalised explanation, such as above, for their statement in lines 219-222: “In the children population, the metabolites involved in (1) histidine metabolism, (2) riboflavin metabolism, (3) pantothenate and CoA biosynthesis and (4) fatty acid biosynthesis were found to change with age in both boys and girls”; why would these different pathways (labelled 1–4 only for the purpose of my feedback) be affected? The “changes” referred to, were these up- or down-regulated? Why? The explanation provided in lines 277-282 is also very nice and it makes sense for the reader. Such explanations for all of their observed trends would enhance the article’s usefulness and impact significantly.

Discussion:
• S4.1: Why do females have higher Trp levels (line 304)?
• S4.1: The authors mention that uric acid is higher in boys than girls (line 315), and this contributes to a higher prevalence of hyperuricemia. Can they, however, speculate on WHY this would be the case (physiology/biochemistry)?

• In S4.1 the authors state that Trp is likely higher in girls (line 303/304, therefore contributing to higher levels of its catabolism products (5-OH-indoleacetic acid). Yet, in S4.3 (lines 368–369), they state that Trp-metabolites (5-methoxytryptophan) were higher in boys. Similarly, line 321 higher acylcarnitine levels in males, but in line 402, these are reported as higher in girls… These statements seem in contrast with each other…

• S4.3: Why would arachidonic acid metabolites need to be higher in females (lines 409–412), is there a physiological need perhaps?
• S4.3: With increased 8-OH-7-methylguanine detected in men (line 414–420), does this mean men is more prone to develop tumors/cancer?

• General: The authors need to include more reasoning (why?) in their discussion(s). Simply stating that a compound is increased in males vs females and that this corresponds with previous results isn’t sufficient. [Please also see my ‘general comment’ for Sections 3 — these explanations can be included/moved to the discussion instead.] Such explanations for all of their observed trends would enhance the article’s usefulness and impact significantly.

Annotated reviews are not available for download in order to protect the identity of reviewers who chose to remain anonymous.

Reviewer 2 ·

Basic reporting

The manuscript is overall well written, despite some typographical and grammatical errors. The cited litterature is appropriate and the article is well structured.
Raw data are not shared. The peaktables used for statistical analyses should at least be accessible to readers.

Experimental design

- Regarding metabolite identification, is there MS/MS information for all metabolites displayed in the table 1? How are the MS/MS scores calculated? Are these the "scores" from the table S1? It is important to know which reference MS² spectra from the public database have been used (there are often several spectra per compound), and the instrument with which they were acquired should be specified. Figure S3 gives information on only a few compounds and it is difficult to compare spectra based on images alone. For important compounds, it is necessary to acquire reference MS² spectra in the laboratory with authentic standards.
In Table S1, some metabolites are detected in other forms than the protonated one. Some like Tyrosyl-valine or glutamine should be detected in the protonated form. Is there an explanation for this? On the other hand, glutamine appears twice in the table (in the L and D forms) and at two different masses and two very different retention times. This is very unlikely given the chromatographic column used.
To summarize, please carefully check the data in table S1 and validate the identification of metabolites of biological relevance using authentic standards.
- Regarding urine sample preparation, I understand that urine samples were not normalized to correct from diuresis variations. This should be discussed in the manuscript.
- Why was the MS detection only achieved using the positive ionization mode?
- Were the samples from each cohort analyzed all at once or in multiple batches?

Validity of the findings

This study could be of interest for those who are involved in research activities dealing with metabolism, physiology and biomarker discovery using metabolomics. However, the validity of the findings is highly dependent on the strength of the metabolite identification data.

Additional comments

- Table 2 and Figure 2: which metabolic pathway ontology was used by the authors? This information should be provided as footnote.
- Figure 2: please indicate the software used for designing this figure in the legend or in the methods section.
- Figure 3c and 5b: what is “num”? This should be clarified.
- Figure 4: the color code is missing.
- Figure 5a: the color code is missing. Are there any statistical differences? These two figures include data that were acquired in two separate experiments. Did the authors normalize the values before? How did they check that the signal intensities are comparable? This should be clarified at least in the figure legend.
- Table S4: what are the “factors”?
- The peak tables used for statistical analyses should be shared.

---

## Round 0.2 · Minor Revisions

Please take note of the comments to clarify the remaining questions.

Reviewer 2 ·

Basic reporting

No particular comments

Experimental design

- Concerning section 2.2: urine sample preparation: is the QC sample common to both cohorts or on the contrary, is there a QC sample for the pediatric cohort and another for the adult cohort? This should be clarified in the manuscript. The clustering observed in Figure S1a suggests that there is only one QC, common to both cohorts.
- Concerning section 2.5: Feature annotation and metabolite identification: the score value is calculated from mass error, isotope similarity, and fragmentation similarity. However, it is not indicated with which formula this score is obtained. Why is the maximum value of the score set to "60"?

Validity of the findings

No particular comments

Additional comments

In the excel file of tables S1-S8, there is a tab "sheet 1" which is not described, and which contains values which are not the same as those of table S1a. What does this tab correspond to?

Annotated reviews are not available for download in order to protect the identity of reviewers who chose to remain anonymous.

---

## Round 0.3 · accepted · Accept

Thanks for sufficiently addressing the final revisions suggested by the reviewer.